# Learning to Distinguish: Behavior Gap Optimization for Goal-Conditioned Policy Learning

## Abstract

Goal-conditioned reinforcement learning (GCRL) trains agents to accomplish a wide variety of tasks by optimizing goal-conditioned policies to achieve desired goals. However, a critical challenge in GCRL is the insufficient separation between the value estimates of optimal and suboptimal actions, a phenomenon we refer to as the Insufficient Behavior Gap, which can significantly degrade policy performance. To address this issue, we propose Behavior Gap Optimization Goal-Conditioned RL (BG2RL), a method that explicitly maximizes this gap through a contrastive optimization framework. Specifically, BG2RL samples reachable future states as target goals, which are considered positive examples, and strategically selects challenging, unachieved states from other trajectories as non-target goals, regarded as negative examples. By maximizing the value disparity between actions leading to these distinct outcomes, BG2RL learns a more discriminative value function and a more robust policy. Theoretical analysis shows that enlarging the policy gap between target and non-target goals directly tightens the suboptimality bound, providing a formal guarantee for the effectiveness of our contrastive objective. Finally, extensive experiments on challenging MuJoCo-based robotic manipulation tasks demonstrate that BG2RL significantly outperforms existing GCRL baselines in terms of success rate and exhibits more stable performance in environments with added obstacles, validating its robustness for goal-directed policy learning.

## 1 Introduction

Goal-conditioned reinforcement learning (GCRL) (Andrychowicz et al., 2017; Fang et al., 2019; Ding et al., 2019; Yang et al., 2022) trains agents to accomplish desired goals and has proven effective in various fields, including robotic manipulation (Li et al., 2023; Nair & Finn, 2020), task sequencing (Nasiriany et al., 2019), and interactive games (Anschel et al., 2017). Like conventional reinforcement learning approaches (Wang et al., 2024), GCRL incorporates states and desired goals as inputs to formulate policies. Both paradigms depend on bootstrapping, a process that iteratively refines value estimates using earlier approximations. However, this bootstrapping process can propagate and amplify estimation errors when the value functions for different goals are similar, making it difficult to distinguish between optimal paths to target goals and suboptimal paths to non-target goals. Nevertheless, the smaller gap between optimal and suboptimal policies, due to intertwined signals from target and non-target goals, can lead to policy confusion and impede the efficient policy optimization

The discrepancy between estimated and optimal action values notably impairs the performance of goal-conditioned policies. A viable strategy to address this is the Increasing Action Gap (IAG) method (Bellemare et al., 2016; Zhang et al., 2022; Eysenbach et al., 2021), which expands the separation between optimal and suboptimal actions. IAG utilizes a consistent Bellman operator to diminish the values of suboptimal actions while upholding the target policy. This expansion facilitates improved action choices and accelerates convergence in discrete environments. However, IAG is inherently limited to DQN-based algorithms in discrete action spaces, as it requires computing Q-values for all available actions to identify and suppress the second-best action. In continuous action spaces, where the action set is infinite, this exhaustive evaluation becomes computationally

intractable. Therefore, for continuous control tasks, suboptimal values or policies must be dynamically tailored relative to the learned optimal ones through alternative mechanisms to effectively broaden the policy gap.

In addition to action gap expansion, contrastive guided policy learning offers another avenue to minimize value deviations from optimality. Contrastive Representation Learning (CRL) (Yuan et al., 2022; Zheng et al., 2023; Wang et al., 2023) reconceptualizes GCRL as a representation learning task, designating state goal pairs from the same trajectory as positive examples and those from separate trajectories as negative examples. CRL eschews traditional RL elements, such as value functions and rewards, in favor of a contrastive loss to direct representation acquisition. Although this method theoretically supports policy enhancement through contrastive principles (Mesnard et al., 2021), it demands substantial interactions with the environment to produce informative samples. Furthermore, the omission of explicit reward signals restricts its utility, especially in multi-goal scenarios where task specific incentives are essential for achieving varied objectives. As a result, CRL techniques often yield only approximate policy improvements and may constrain exploratory behavior.

In this paper, Behavior Gap Optimization Goal-Conditioned RL (BG2RL) is introduced—a method designed to diminish the gap between learned and optimal policies by adaptively enlarging the separation between action values conditioned on target and non-target goals. Target goals, which enable gradual task accomplishment and represent efficient paths to success, are contrasted with non-target goals that simulate inefficient or failed trajectories where desired outcomes are not achieved. Specifically, non-target goals are strategically sampled to initially minimize the action value gap with their corresponding target goals, ensuring that the most challenging negative samples—those that are hardest to distinguish from successful paths—are selected for training. This minimal initial gap is then systematically maximized through policy optimization based on the goal-conditioned action value difference, employing a mechanism analogous to contrastive learning Mesnard et al. (2021); Forney et al. (2017) where target goals are treated as positive samples and non-target goals as negative samples. By enlarging this action value gap through targeted optimization, the suboptimality bound of learned policies relative to optimal ones is provably reduced by BG2RL, as demonstrated in our theoretical analysis. This approach enables more robust goal-directed policy learning, with significantly higher task success rates being achieved compared to existing methods.

## 2 RELATED WORKS

Our proposed method introduces a goal-conditioned reinforcement learning approach that enhances the policy gap (Gong et al., 2024; Yalcinkaya et al., 2024; Qiu et al., 2023), drawing inspiration from action gap increasing techniques (Bellemare et al., 2016; Zhang et al., 2022) and contrastive policy learning (Mesnard et al., 2021; Forney et al., 2017). These mechanisms facilitate the differentiation between optimal and suboptimal policies, thereby promoting robust policy improvement.

**Action Gap Increasing** The Increasing Action Gap (IAG) method (Bellemare et al., 2016) enhances policy learning efficiency by widening the disparity between optimal and suboptimal action values, where suboptimal values are derived from the second-best actions in discrete tasks. This is achieved through a consistent Bellman operator that incorporates both optimal and suboptimal action values. Similarly, Clipped Advantage Learning (CAL) (Zhang et al., 2022) adaptively modulates advantage values to balance substantial action gaps with rapid convergence, mitigating mismatches in optimal actions and yielding strong performance on reinforcement learning benchmarks. Nonetheless, both IAG and CAL are limited to discrete control tasks, as suboptimal value estimation relies on selecting the second-largest action value.

**Contrastive Learning** Contrastive Unsupervised Reinforcement Learning (CURL) (Laskin et al., 2020) leverages contrastive objectives to bolster representation learning (Yin et al., 2024) by contrasting augmented versions of the same observation, thereby improving the robustness and efficiency of state representations. Contrastive Reinforcement Learning (CRL) (Mesnard et al., 2021; Forney et al., 2017) employs a contrastive framework to maximize the separation between goal-conditioned and non-target policies, functioning as a representation learning paradigm that eschews explicit action-value learners for policy evaluation. Furthermore, Contrastive Preference Learning (CPL) (Hejna et al., 2023) integrates contrastive learning with human feedback to support preference-based decision-making without conventional reinforcement learning components. Although these approaches demonstrate stability in targeted tasks, their policy training often omits

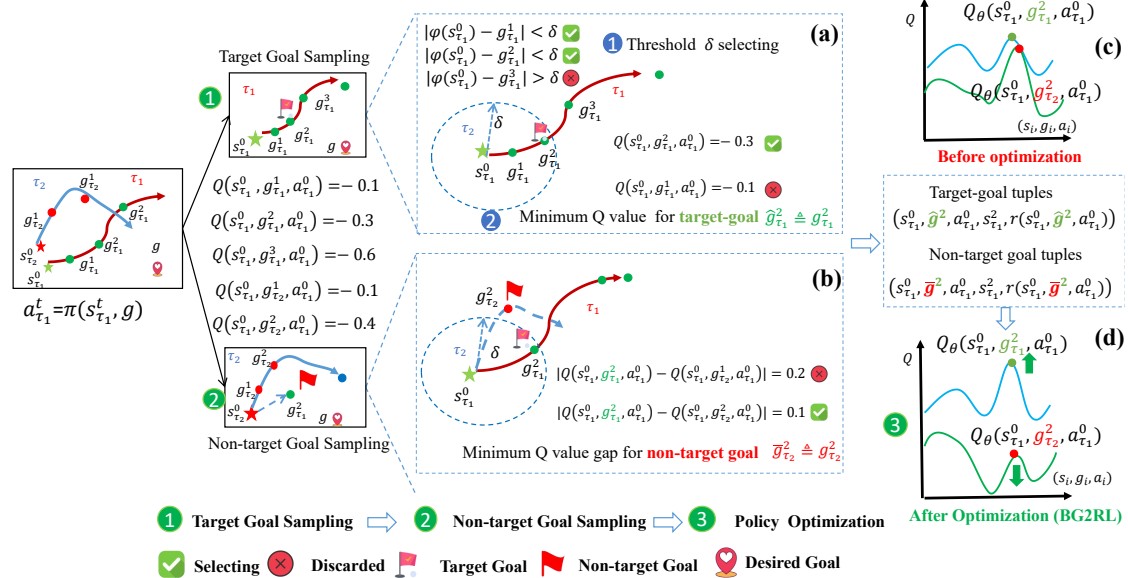

Figure 1: The framework of our proposed method for enhancing the policy gap in GCRL.

reward signals in action-value estimation, which can impair goal achievement. For example, CRL (Mesnard et al., 2021) exhibits reduced success rates in multi-goal scenarios. Consequently, contrastive objectives should be integrated with reward-based policy evaluation to foster improved policy refinement.

## 3 BACKGROUND

Goal-conditioned reinforcement learning, abbreviated as GCRL, is a framework for training an agent to achieve specific goals via environmental interactions. It operates in an infinite-horizon Markov Decision Process, denoted as MDP, tuple $\langle \mathcal{S}, \mathcal{G}, \mathcal{A}, \mathcal{R}, \mathcal{P}, \gamma \rangle$, where $\mathcal{S}$ is the state space describing environment conditions, $\mathcal{G}$ the goal space describing possible objectives, $\mathcal{A}$ the action space, $\mathcal{P}$ the transition probabilities from state $s_t$ to $s_{t+1}$ under action $a_t$, $\mathcal{R}$ the reward function, and $\gamma \in [0, 1)$ the discount factor balancing immediate and future rewards. The goal space $\mathcal{G}$ derives from $\mathcal{S}$ as $\mathcal{G} = \varphi(\mathcal{S})$, with $\varphi$ a function defined by the environment that maps a state $s_t$ to an achieved goal $\varphi(s_t)$.

Given state $s_t$ and desired goal $g$, the agent selects $a_t = \pi(s_t, g)$ via goal-conditioned policy $\pi$, learned to reach $g$. For instance, in a robotic manipulation task, the state $s_t$ represents the current configuration of the robot, such as its joint angles and velocities; the desired goal $g$ specifies the target position, like the coordinates of an object to grasp; and the achieved goal $\varphi(s_t)$ maps the state to the current outcome, such as the actual position of the robot's end-effector. The achieved goal $\varphi(s_t)$ directly relates to the state, reflecting progress toward $g$. The reward $r(s_t, g, a_t)$ is given as follows:

$$ r(s_t, g, a_t) = \begin{cases} 0 & \text{if } d(\varphi(s_{t+1}), g) < \delta \\ -1 & \text{otherwise} \end{cases} $$

where $d$ represents a distance metric and $\delta$ denotes a success threshold set at 0.05, a reward of 0 signifies goal achievement, whereas -1 penalizes delays. The objective in goal-conditioned reinforcement learning (GCRL) is to maximize the expected return: $\mathcal{J}(\pi) = \mathbb{E}_{g \sim \rho_g, \tau \sim d^\pi(\cdot|g)} [\sum_{t=0}^{\infty} \gamma^t r(s_t, g, a_t)]$, where $d^\pi(\tau|g)$ is the trajectory distribution induced by the policy for goal $g$.

## 4 METHOD

Our method improves policy learning in Goal-Conditioned Reinforcement Learning (GCRL) by explicitly maximizing the policy gap—defined as the difference in action-value estimates between an optimal policy that successfully achieves a target goal and a suboptimal policy that fails to do so. The optimal policy is approximated using relabeled target goals $\hat{g}_{\tau_i}^t$ derived from the current trajectory $\tau_i$ for state $s_{\tau_i}^t$, representing successful goal-reaching paths. On the other hand, the suboptimal policy employs relabeled non-target goals $\bar{g}_{\tau_i}^t$ from other trajectories $\tau_{j \neq i}$, simulating failure for the same state $s_{\tau_i}^t$.

### 4.1 FRAMEWORK

Figure 1 illustrates the framework of our proposed method, which integrates target and non-target goal sampling with policy optimization through action-value gap maximization. As shown in the left panel, we start with two trajectories, $\tau_1 = \{s_{\tau_1}^0, s_{\tau_1}^1, s_{\tau_1}^2, s_{\tau_1}^3\}$ and $\tau_2 = \{s_{\tau_2}^0, s_{\tau_2}^1, s_{\tau_2}^2\}$, collected under the behavior policy. Both trajectories share the same goal $g$, drawn from a task-specific distribution, but begin from different initial states. From these trajectories, we compute the corresponding goal-conditioned action values for five samples as $Q(s_{\tau_1}^0, g_{\tau_1}^1, a_{\tau_1}^0) = -0.1$, $Q(s_{\tau_1}^0, g_{\tau_1}^2, a_{\tau_1}^0) = -0.3$, $Q(s_{\tau_1}^0, g_{\tau_1}^3, a_{\tau_1}^0) = -0.6$, $Q(s_{\tau_1}^0, g_{\tau_2}^1, a_{\tau_1}^0) = -0.1$, $Q(s_{\tau_1}^0, g_{\tau_2}^2, a_{\tau_1}^0) = -0.4$, and then sample both target and non-target goals based on the learned action values to enhance the action value gap for policy optimization.

In particular, to enhance the policy gap at state $s_{\tau_1}^0$, our method differentiates between target goals $\hat{g}_{\tau_1}^2$, which lead to successful outcomes, and non-target goals $\hat{g}_{\tau_2}^2$, which simulate failure. For instance, in the target goal sampling panel, $g_{\tau_1}^2$ from trajectory $\tau_1$ serves as the target goal for state $s_{\tau_1}^0$, chosen to minimize the Q-value under a distance constraint $\delta$. This ensures non-sparse rewards and encourages the agent to explore different goals. On the other hand, in the non-target goal sampling panel, $g_{\tau_2}^2$ from trajectory $\tau_2$ is selected as the non-target goal because it exhibits the smallest action-value gap with $g_{\tau_2}^1$ for state $s_{\tau_1}^0$. Although the figure illustrates the goal construction process only for state $s_{\tau_1}^0$ in trajectory $\tau_1$, a similar process applies to trajectory $\tau_2$. Finally, policy optimization maximizes the action value gap between target and non-target goals, as shown in panel (d) of Figure 1. Here, the target action value is increased, while the non-target action value is decreased, which enables more stable policy learning.

### 4.2 TARGET GOAL SAMPLING

The target goal $\hat{g}_{\tau_i}^t$ corresponds to a reachable goal that the agent can achieve with minimal unnecessary steps, ultimately leading to task success. It is sampled from the current trajectory for a specific state. Given a trajectory $\tau_i = \{(s_{\tau_i}^t, g, a_{\tau_i}^t, s_{\tau_i}^{t+1}, r(s_{\tau_i}^t, g, a_{\tau_i}^t))\}$, the achieved goal $\hat{g}_{\tau_i}^t = \varphi(s_{\tau_i}^{t+k})$ for $k \geq 1$ in $\tau_i$ can be viewed as a relabeled desired goal for state $s_{\tau_i}^t$ in trajectory $\tau_i$, where $\varphi$ maps states to the goal space $\mathcal{G} \subseteq \mathcal{S}$. The target goal $\hat{g}_{\tau_i}^t$ is chosen to be the most reachable goal from the current state $s_{\tau_i}^t$, which minimizes the unnecessary steps to achieve task success. The target goal sampling criterion is defined as:

$$\hat{g}_{\tau_i}^t = \arg \min_{\hat{g}_{\tau_i}^t \in \tau_i} Q_\theta(s_{\tau_i}^t, \hat{g}_{\tau_i}^t, a_{\tau_i}^t) \quad \text{s.t.} \quad \left| \hat{g}_{\tau_i}^t - \varphi(s_{\tau_i}^t) \right|_1 \leq \delta \tag{1}$$

where $\hat{g}_{\tau_i}^t$ target goal is selected by minimizing the action value $Q_\theta(s_{\tau_i}^t, \hat{g}_{\tau_i}^t, a_{\tau_i}^t)$, subject to a distance constraint $\left| \hat{g}_{\tau_i}^t - \varphi(s_{\tau_i}^t) \right|_1 \leq \delta$, ensuring that the goal is within a reasonable range to avoid sparse rewards. Goals with higher action values are discarded, and the goal with the minimum action value, representing the most efficient path to success, is selected. This method balances exploration and exploitation by focusing on the most promising goals.

Figure 1 illustrates this target goal sampling process. For example, if $g_{\tau_1}^3$ from trajectory $\tau_1$ is farther from state $s_{\tau_1}^0$ than the threshold $\delta$, it is discarded. Besides, $g_{\tau_1}^1$, which is too close to $s_{\tau_1}^0$, is discarded due to its higher action value, indicating a less favorable goal for exploration. Notably, In our environment, the agent receives a reward of -1 for each step taken and 0 upon reaching the goal. Consequently, the action-value function at the initial state reflects the expected number of steps to the goal: lower (more negative) values indicate longer paths to more distant goals, while

higher (closer to zero) values indicate shorter paths to nearby goals. The goal $g_{\tau_1}^2$ with the minimum action value is selected, as it strikes the best balance between reaching a goal efficiently and avoiding sparse rewards, making it the most suitable target for the agent to pursue.

## 4.3 Non-target Goal Sampling

The non-target goal $\bar{g}_{\tau_i}^t$ corresponds to an unreachable goal, and it is sampled from a different trajectory $\tau_j$ for state $s_{\tau_i}^t$. Specifically, for state $s_{\tau_i}^t$ from trajectory $\tau_i$, the non-target goal $\bar{g}_{\tau_i}^t$ is sampled from a distinct trajectory $\tau_j$ (where $j \neq i$), forming the tuple $(s_{\tau_i}^t, \bar{g}_{\tau_i}^t, a_{\tau_i}^t, s_{\tau_i}^{t+1}, r(s_{\tau_i}^t, \bar{g}_{\tau_i}^t, a_{\tau_i}^t))$. These non-target goals are considered "unreachable" for state $s_{\tau_i}^t$, which implies that following the corresponding actions will lead to suboptimal or failed outcomes. The formal criterion for selecting the non-target goal is:

$$\bar{g}_{\tau_i}^t = \arg \min_{\bar{g}_{\tau_i}^t \in \tau_j, j \neq i} \left| Q_\theta(s_{\tau_i}^t, \hat{g}_{\tau_i}^t, a_{\tau_i}^t) - Q_\theta(s_{\tau_i}^t, \bar{g}_{\tau_i}^t, a_{\tau_i}^t) \right| \tag{2}$$

where $\bar{g}_{\tau_i}^t$ is the non-target goal sampled from trajectory $\tau_j$, with $j \neq i$, meaning it comes from a different trajectory than the one currently being processed. The non-target goal is selected from the samples that characterized by minimizing the absolute difference between these two action values $Q_\theta(s_{\tau_i}^t, \hat{g}_{\tau_i}^t, a_{\tau_i}^t)$ and $Q_\theta(s_{\tau_i}^t, \bar{g}_{\tau_i}^t, a_{\tau_i}^t)$, ensuring that the non-target goal closely resembles the target goal in terms of the trajectory but leads to a suboptimal result.

Figure 1(b) illustrates the non-target goal sampling mechanism. For a given state $s_{\tau_1}^0$ in trajectory $\tau_1$, the candidate non-target goals are $g_{\tau_2}^1$ and $g_{\tau_2}^2$, which correspond to goals in a different trajectory $\tau_2$. The non-target goal is selected by minimizing the action value gap between the target goal $\hat{g}_{\tau_1}^2$ and the candidate non-target goals $g_{\tau_2}^1$ and $g_{\tau_2}^2$. In this example, $g_{\tau_2}^2$ is selected because it exhibits the smallest action value gap compared to $g_{\tau_2}^1$. This ensures that the selected non-target goal is relevant to the current state but leads to a suboptimal outcome, thus helping the agent to differentiate between successful goal-reaching behaviors and those that lead to failure.

## 4.4 Policy Gap Optimization

The goal of our approach is to maximize the action-value gap between target and non-target transitions, thereby enhancing the policy gap. For a given state $s_{\tau_i}^t$, target goal $\hat{g}_{\tau_i}^t$, and non-target goal $\bar{g}_{\tau_i}^t$, we learn the corresponding goal-conditioned action-value functions, $Q_\theta(s_{\tau_i}^t, \hat{g}_{\tau_i}^t, a_{\tau_i}^t)$ and $Q_\theta(s_{\tau_i}^t, \bar{g}_{\tau_i}^t, a_{\tau_i}^t)$, using the Deep Deterministic Policy Gradient (DDPG) algorithm. The action $a_{\tau_i}^t$ is determined by the behavior policy $\pi_\phi(s_{\tau_i}^t, g)$, which generates an action based on the current state and goal. The objective is to maximize the difference between the action values for the target goal and the non-target goal, as defined by the following loss function:

$$\mathcal{J}_\phi(s_{\tau_i}^t, \hat{g}_{\tau_i}^t, \bar{g}_{\tau_i}^t) = \max_\phi \mathbb{E}_{s_{\tau_i}^t, \hat{g}_{\tau_i}^t, \bar{g}_{\tau_i}^t \sim \mathcal{D}} \left[ Q_\theta(s_{\tau_i}^t, \hat{g}_{\tau_i}^t, a_{\tau_i}^t) - \alpha \mathcal{I}(s_{\tau_i}^t, \hat{g}_{\tau_i}^t, \bar{g}_{\tau_i}^t) Q_\theta(s_{\tau_i}^t, \bar{g}_{\tau_i}^t, a_{\tau_i}^t) \right] \tag{3}$$

where $\alpha$ is a hyper-parameter, the first term corresponds to the goal-conditioned action-value function for the target goal, $Q_\theta(s_{\tau_i}^t, \hat{g}_{\tau_i}^t, a_{\tau_i}^t)$, which is trained by maximizing the action-value estimates through standard reinforcement learning techniques. This ensures that the agent learns to assign higher action values to actions leading to successful goal achievements. The second term involves the action-value function for the non-target goal, $Q_\theta(s_{\tau_i}^t, \bar{g}_{\tau_i}^t, a_{\tau_i}^t)$, and is conditionally minimized using the indicator function $\mathcal{I}$. Specifically, the indicator function $\mathcal{I}(s_{\tau_i}^t, \hat{g}_{\tau_i}^t, \bar{g}_{\tau_i}^t)$ prevents over-penalizing non-target action values, helping stabilize training by activating only when the non-target action value exceeds the target action value. The indicator function is defined as:

$$\mathcal{I}(s_{\tau_i}^t, \hat{g}_{\tau_i}^t, \bar{g}_{\tau_i}^t) = \begin{cases} 1, & \text{if } Q_\theta(s_{\tau_i}^t, \hat{g}_{\tau_i}^t, a_{\tau_i}^t) < Q_\theta(s_{\tau_i}^t, \bar{g}_{\tau_i}^t, a_{\tau_i}^t), \\ 0, & \text{otherwise.} \end{cases} \tag{4}$$

where $\mathcal{I}(s_{\tau_i}^t, \hat{g}_{\tau_i}^t, \bar{g}_{\tau_i}^t) = 1$ when the target goal's action value is lower than the non-target goal's action value, i.e., when $Q_\theta(s_{\tau_i}^t, \hat{g}_{\tau_i}^t, a_{\tau_i}^t) < Q_\theta(s_{\tau_i}^t, \bar{g}_{\tau_i}^t, a_{\tau_i}^t)$. This design serves a dual purpose: first, it further amplifies the action-value gap when the model already correctly assigns lower values

to target goals which require more steps, thereby reinforcing the desired policy differentiation; second, it avoids penalizing cases where the ordering is reversed, which may result from noisy value function estimates during early training stages. By selectively applying the penalty only when the correct ordering is present, our method robustly enhances the policy gap while maintaining training stability.

The objective in Eqn. 3 drives the agent to distinguish between successful and failed tasks by maximizing the action-value gap between target and non-target goals. This approach not only helps the agent avoid suboptimal paths but also strengthens decision-making by reinforcing successful trajectories. By leveraging contrastive learning, the agent learns to better differentiate between effective and ineffective actions, thus reducing action-value estimation errors. This enhanced policy optimization is further validated by the theoretical analysis presented in Theorem 1, which demonstrates the benefits of widening the gap between successful and failed trajectories.

**Theorem 1** (Policy Suboptimality via Gap Maximization). *Let $\pi_\phi$ denote the policy optimized to maximize the action-value gap between target goal $\hat{g}$ and non-target goal $\bar{g}$, where $\hat{g}$ encourages efficient goal-reaching and $\bar{g}$ simulates failure cases. Under Assumptions 1–3, for any initial state $s_0$ and target goal $\hat{g}$, the suboptimality of $\pi_\phi$ is bounded by:*

$$V^*(s_0, \hat{g}) - V^{\pi_\phi}(s_0, \hat{g}) \leq \frac{4\epsilon_Q}{1 - \gamma} + \frac{2R_{\max}}{(1 - \gamma)^2} - \frac{\Delta_{\min}}{1 - \gamma}, \quad (5)$$

*where $\epsilon_Q$ bounds the uniform Q-approximation error, $R_{\max}$ is the maximum absolute reward magnitude, $\gamma$ is the discount factor, and $\Delta_{\min} = \mathbb{E}_{s,a \sim \pi_\phi}[Q^{\pi_\phi}(s, \hat{g}, a) - Q^{\pi_\phi}(s, \bar{g}, a)]$ is the expected policy gap under the discounted state-action visitation distribution induced by $\pi_\phi$ for goal $\hat{g}$. The bound strictly decreases with $\Delta_{\min}$ and remains non-negative for all feasible gaps.*

Theorem 1 establishes a finite-sample suboptimality bound for goal-conditioned policies trained with contrastive goal sampling. It demonstrates that the performance gap between the learned policy and the optimal policy decreases linearly as the expected action-value gap, $\Delta_{\min}$, between target and non-target goals increases. In essence, the theorem provides a theoretical guarantee that maximizing the policy gap is not only an intuitive approach but also provably effective in enhancing goal-reaching performance.

## 5 Experiments

Our experiments were conducted on a four-machine cluster equipped with one Intel Xeon(R) W-2255 CPU and one NVIDIA GeForce RTX 3090 GPU per machine. Each task was evaluated using 20 rollouts, with each agent trained for 50 epochs using a batch size of 256. To ensure robustness, experiments were run across five random seeds, and results are reported as mean success rates with standard deviations. All other hyperparameters were set to match those in (Andrychowicz et al., 2017). We compared our method, BG2RL, against the following baselines: **HER** (Hindsight Experience Replay) (Andrychowicz et al., 2017), a foundational goal-conditioned reinforcement learning method for addressing sparse reward issues through relabeling; **IAG** (Increasing Action Gap) (Bellemare et al., 2016), which explicitly increases the gap between optimal and suboptimal action values; and **CRL** (Contrastive Reinforcement Learning) (Mesnard et al., 2021; Forney et al., 2017), which reformulates goal-conditioned RL as representation learning based on contrastive principles. For **IAG**, we adapted it to continuous action spaces by adding clipped random noise to actions suggested by the target network, thereby generating suboptimal actions. While IAG directly maximizes action gaps, HER and CRL achieve related benefits implicitly through relabeling and contrastive objectives, respectively. We evaluated our proposed goal-conditioned method using MuJoCo-based control tasks, specifically robotic manipulation environments including FetchReach, FetchPush, FetchPickAndPlace, FetchSlide and its obstacles version `FetchPush_Obs FetchSlide_Obs`, and Point-based tasks, as illustrated in Fig. 6 in Appendix. These tasks feature continuous state and action spaces, with sparse rewards and episode lengths of up to 50 steps.

### 5.1 Evaluation Performance

Our method is evaluated on Fetch and Ant based robotic tasks, with performance measured by the success rate in manipulating the robot to reach specific goals. The results in Fig-

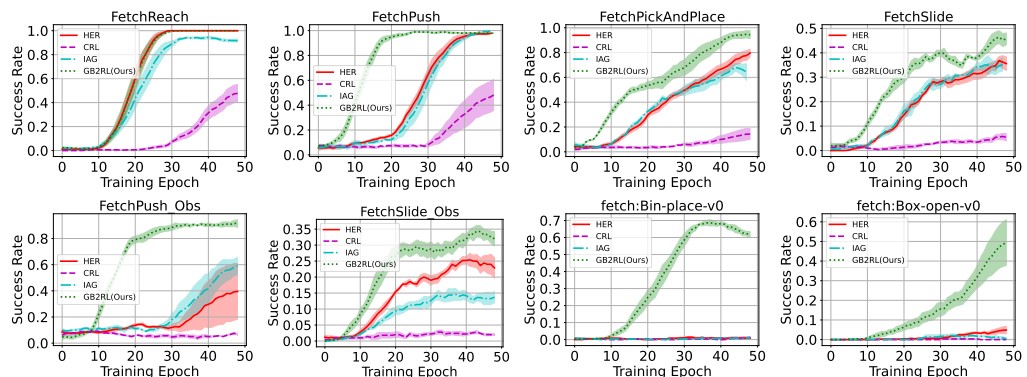

Figure 2: The evaluation results compared with baselines on success rate.

Table 1: Average success rate of different algorithms on various environments.

| Algorithm | Environment | | | | | | | |
|---|---|---|---|---|---|---|---|---|
| | Reach | Push | Pick | Slide | push-v2 | slide-v2 | Binplace | Boxopen |
| HER | 1.0000 | 0.9762 | 0.7857 | 0.3524 | 0.3952 | 0.2286 | 0.0086 | 0.0486 |
| CRL | 0.5143 | 0.1429 | 0.1048 | 0.0143 | 0.0667 | 0.0200 | 0.0095 | 0.0000 |
| IAG | 0.9086 | 0.8619 | 0.5714 | 0.1333 | 0.5905 | 0.1371 | 0.0095 | 0.0048 |
| GB2RL(Ours) | 1.0000 | 0.9952 | 0.8524 | 0.4524 | 0.9143 | 0.3200 | 0.6171 | 0.4943 |

ure 2 and Tab. 1 demonstrate that our method achieves the most stable performance compared to the baselines, including HER (Andrychowicz et al., 2017), Where `FetchReach`, `FetchPush`, `FetchPickAndPlace`, `FetchSlide`, `FetchPush_Obs`, `FetchSlide_Obs`, `fetch:Bin-place-v0`, and `fetch:Box-open-v0` are abbreviated as `Reach`, `Push`, `Pick`, `Slide`, `push-v2`, `slide-v2`, `Binplace`, and `Boxopen` respectively. This stability arises from the enlarged gap in our learned goal-conditioned action values between target and non-target goals, reducing ambiguity and estimation errors.

In addition, compared to IAG (Bellemare et al., 2016), our method exhibits superior success rates and sample efficiency. This advantage stems from IAG's approach to increasing action gaps being less suitable for continuous tasks, where suboptimal actions are generated by adding random noise to policy-suggested actions, thereby hindering learning efficiency. Compared to CRL (Mesnard et al., 2021), CRL attains higher success rates than HER in Ant- and Point-based tasks but lower rates in Fetch-based tasks. This discrepancy results from CRL's omission of explicit desired goals and reward signals, relying instead on contrastive objectives for representation learning to quantify state-goal similarities, which limits its effectiveness in multi-goal environments requiring task-specific incentives. Overall, the results indicate that our method's policy gap enlargement enables stable performance across diverse tasks, with enhanced sample efficiency and success rates.

## 5.2 ACTION VALUE AND POLICY GAP INVESTIGATION

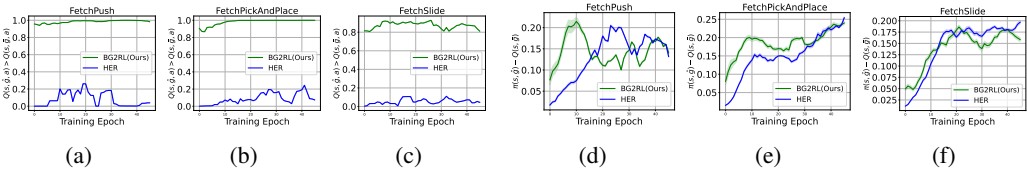

Figure 3: The action value and policy difference in target and non-target goal-conditioned samples.

Our method maximizes the policy gap by optimizing both goal-conditioned and non-target action values. To investigate the policy differentiation between goal-conditioned and non-target scenarios,

we analyze the corresponding action value differences and policy gaps estimated using L1 norm distance, as presented in Figure 3.

To examine the differences in action values, Fig. 3 (a-c) shows the percentage of times the target goal-conditioned action values exceed the non-target goal-conditioned action values throughout training for the three environments. In contrast to HER, BG2RL consistently maintains a high percentage, indicating that its policy consistently favors actions that align with the target goal. Specifically, BG2RL demonstrates a steady increase in this percentage across all environments, reflecting the model's ability to distinguish target goals from non-target ones. This result validates the effectiveness of BG2RL's contrastive optimization framework, which explicitly encourages better goal-directed behavior compared to the more ambiguous actions learned by HER.Fig. 3 (d-e) compares the policy difference between BG2RL and HER across the three environments. The figure shows that BG2RL consistently maintains a larger policy gap, particularly in the early stages of training, indicating that our method effectively distinguishes between optimal and suboptimal actions right from the beginning. In contrast, HER shows a smaller gap, with the policy difference gradually increasing but never reaching the level of BG2RL. This highlights the early advantage of BG2RL in learning more goal-directed policies, which is crucial for improving long-term performance.

## 5.3 CRITIC SURFACE VISUALIZATION

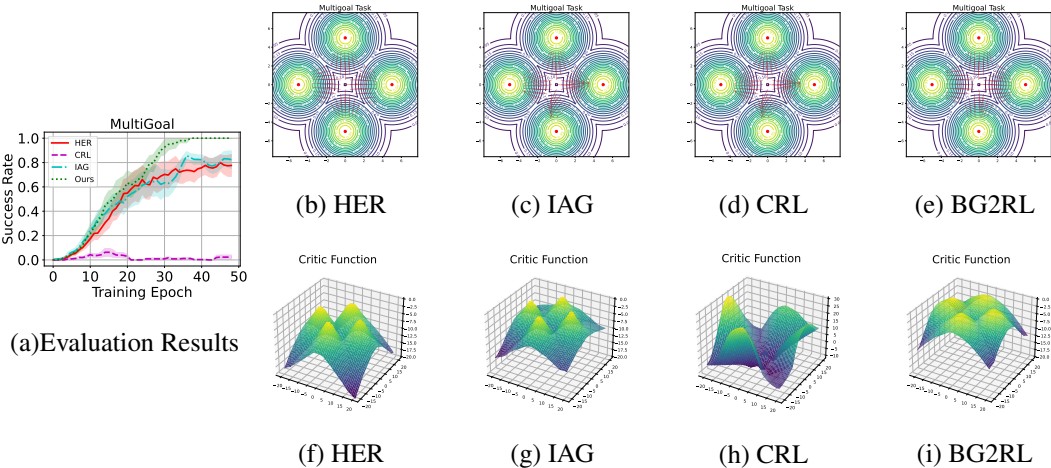

Figure 4: The top row (a) show the evaluation results on multi-goal task, (b–e) illustrates policy distributions for a multi-goal task with goals at $[-5, 0]$, $[5, 0]$, $[0, 5]$, and $[0, -5]$, evaluated via evaluation results.

To investigate the effect of increased action value gaps on the learned critic module, we evaluate policy distributions and critic surfaces in a multi-goal task. The task involves four goals positioned at coordinates $[-5, 0]$, $[5, 0]$, $[0, 5]$, and $[0, -5]$, with the agent initialized at the origin. The desired goal is uniformly sampled from these four positions, and the reward is defined using the Euclidean distance between the state and the selected goal: a reward of $-1$ is given if the distance exceeds a predefined threshold, and $0$ otherwise.

Figure 4 reports success rates over training epochs on the left, showing that our method achieves the highest and most stable performance compared to baselines. The figure also compares policy distributions and critic surfaces for our method against HER (Andrychowicz et al., 2017), IAG (Bellemare et al., 2016), and CRL (Mesnard et al., 2021). The top row (a–d) visualizes policy distributions, where red arrows represent action vectors, with their length and direction indicating magnitude and orientation, respectively. The bottom row (e–h) depicts value function surfaces for each method, except for CRL (Mesnard et al., 2021), which reports the contrastive loss surface instead. The policy distributions in the top row of Figure 4 highlight key differences. Methods incorporating explicit goal guidance, such as HER (Andrychowicz et al., 2017), IAG (Bellemare et al., 2016), and ours, generate actions that consistently direct the agent toward the desired goals. In contrast, the CRL (Mesnard et al., 2021) policy (d) produces directionless actions, underscoring its

limitations in navigating multi-goal environments without task-specific incentives. The critic surfaces in the bottom row further illustrate these distinctions. Our method's surface (i) exhibits four smooth peaks, each precisely aligned with the target goal positions and displaying lower gradient variance, indicating effective goal differentiation and guidance with minimal extraneous steps. The HER (Andrychowicz et al., 2017) surface (f) also shows four peaks but with sharper, less smooth gradients compared to ours. The IAG (Bellemare et al., 2016) surface (g) is overly flat, failing to adequately differentiate states near goals from others, which limits task guidance. The CRL (Mesnard et al., 2021) surface (h) lacks distinct peaks, reflecting its inability to capture the multi-goal structure due to the absence of explicit goal-conditioned rewards and reliance on contrastive representation learning.

## 5.4 ABLATION STUDIES

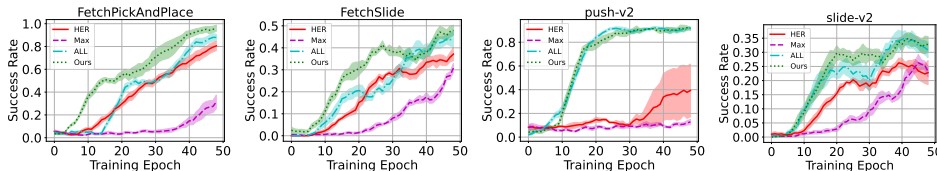

Figure 5: The evaluation results of ablation studies.

Our method enhances goal-conditioned policy learning by maximizing the action value gap between target and non-target goals. To elucidate the operational mechanism of our proposed approach, we conducted ablation studies across various settings, with results illustrated in Figure 5.

First, our method minimizes non-target goal-conditioned values to enhance the policy gap. To assess the necessity of this minimization, we introduce a variant (Max) that modifies the actor training objective to maximize both target and non-target goal-conditioned values, defined as:

$$\max_\phi \mathbb{E}_{(s_{\tau_i}^t, \hat{g}_{\tau_i}^t, a_{\tau_i}^t)} \left[ Q_\theta(s_{\tau_i}^t, \hat{g}_{\tau_i}^t, a_{\tau_i}^t) + Q_\theta(s_{\tau_i}^t, \bar{g}_{\tau_i}^t, a_{\tau_i}^t) \right]$$

where the actor is trained by maximizing these values, denoted as the Max variant. As shown in Figure 5, our method (Ours) achieves a higher success rate than Max. This superiority arises because maximizing non-target values diminishes the policy gap between optimal and suboptimal policies, permitting suboptimal behaviors to interfere with optimal actions.

Second, our method maximizes the policy gap by conditionally minimizing non-target goal values only when they exceed target values. To examine the impact of unconditional minimization, we introduce a variant (ALL) that minimizes all non-target values, defined as:

$$\max_\phi \mathbb{E}_{(s_{\tau_i}^t, \hat{g}_{\tau_i}^t, \bar{g}_{\tau_i}^t)} \left[ Q_\theta(s_{\tau_i}^t, \hat{g}_{\tau_i}^t, a_{\tau_i}^t) - Q_\theta(s_{\tau_i}^t, \bar{g}_{\tau_i}^t, a_{\tau_i}^t) \right]$$

As illustrated in Figure 5, the ALL variant improves sample efficiency in FetchPush but results in unstable performance in FetchPickAndPlace and FetchSlide. These findings highlight that conditional minimization of non-target values, guided by the indicator function, is crucial for stable policy enhancements across diverse tasks.

## 6 CONCLUSIONS

This paper introduces a goal-conditioned reinforcement learning (GCRL) method that enhances policy optimization by maximizing the action-value gap between target and non-target goals (Wu et al., 2022; Lyu et al., 2024). Specifically, our method increases the disparity in action values between reachable target goals and unreachable non-target goals, thereby enhancing the discriminative capacity of the learned policy. Experimental results across various GCRL tasks show that our approach outperforms existing methods aimed at improving policy gaps. However, we observe that in some cases, the action value magnitudes for non-target goals may differ significantly from those of target goals, potentially reducing the intended policy gap and negatively impacting performance. To address this, future work will focus on developing a non-target goal generation mechanism that carefully controls the action value differences between target and non-target goals, further optimizing their contribution to policy gap enhancement.

## LANGUAGE MODEL USAGE STATEMENT

We used a large language model (LLM) to assist with text polishing and grammar checking. The model was used only for improving clarity and correctness of the language. No content, claims, or scientific conclusions were generated or altered by the LLM.

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

# A   THEORETICAL ANALYSIS

## A.1   PROBLEM SETUP AND ASSUMPTIONS

We consider a goal-conditioned Markov decision process (GC-MDP) defined by the tuple $(\mathcal{S}, \mathcal{G}, \mathcal{A}, P, r, \gamma)$, where $\mathcal{S}$ is the state space, $\mathcal{G} \subseteq \mathcal{S}$ is the goal space, $\mathcal{A}$ is the action space, $P(s'|s, a)$ is the transition kernel, and $r : \mathcal{S} \times \mathcal{G} \times \mathcal{A} \rightarrow [-R_{\max}, 0]$ is a sparse reward function (e.g., $-1$ per step until goal reached). The discount factor is $\gamma \in (0, 1)$.

Let $\pi_\phi(a|s, g)$ denote a parameterized goal-conditioned policy, and let $Q^\pi(s, g, a)$ be its action-value function:

$$Q^\pi(s, g, a) = \mathbb{E}_{\tau \sim \pi} \left[ \sum_{t=0}^{\infty} \gamma^t r(s_t, g, a_t) \,\middle|\, s_0 = s, a_0 = a \right].$$

The corresponding value function is $V^\pi(s, g) = \mathbb{E}_{a \sim \pi_\phi(\cdot|s,g)}[Q^\pi(s, g, a)]$, and the optimal value function is $V^*(s, g) = \sup_\pi V^\pi(s, g)$.

We make the following assumptions:

**Assumption 1** (Bounded Rewards). *There exists $R_{\max} > 0$ such that $|r(s, g, a)| \leq R_{\max}$ for all $(s, g, a)$. Consequently, $\|Q^\pi\|_\infty \leq Q_{\max} := R_{\max}/(1 - \gamma)$ for any policy $\pi$.*

**Assumption 2** (Uniform Q-Approximation Error). *The learned Q-function $Q_\theta$ satisfies $\sup_{s,g,a,\pi} |Q_\theta(s, g, a) - Q^\pi(s, g, a)| \leq \epsilon_Q$.*

**Assumption 3** (Policy Gap Lower Bound). *The policy $\pi_\phi$ satisfies:*

$$\mathbb{E}_{s \sim d^{\pi_\phi}(\cdot|\hat{g}), a \sim \pi_\phi(\cdot|s,\hat{g})} [Q^{\pi_\phi}(s, \hat{g}, a) - Q^{\pi_\phi}(s, \bar{g}, a)] \geq \Delta_{\min} > 0,$$

*where $d^{\pi_\phi}(\cdot|\hat{g})$ is the discounted stationary state distribution induced by $\pi_\phi$ under goal $\hat{g}$, and $\hat{g}, \bar{g}$ are the target and non-target goals sampled by the algorithm.*

## A.2   MAIN RESULT

**Theorem 1** (Policy Suboptimality via Gap Maximization). *Let $\pi_\phi$ denote the policy optimized to maximize the action-value gap between target goal $\hat{g}$ and non-target goal $\bar{g}$, where $\hat{g}$ encourages efficient goal-reaching and $\bar{g}$ simulates failure cases. Under Assumptions 1–3, for any initial state $s_0$ and target goal $\hat{g}$, the suboptimality of $\pi_\phi$ is bounded by:*

$$V^*(s_0, \hat{g}) - V^{\pi_\phi}(s_0, \hat{g}) \leq \frac{4\epsilon_Q}{1 - \gamma} + \frac{2R_{\max}}{(1 - \gamma)^2} - \frac{\Delta_{\min}}{1 - \gamma}, \tag{6}$$

*where $\epsilon_Q$ bounds the uniform Q-approximation error, $R_{\max}$ is the maximum absolute reward magnitude, $\gamma$ is the discount factor, and $\Delta_{\min} = \mathbb{E}_{s,a \sim \pi_\phi}[Q^{\pi_\phi}(s, \hat{g}, a) - Q^{\pi_\phi}(s, \bar{g}, a)]$ is the expected policy gap under the discounted state-action visitation distribution induced by $\pi_\phi$ for goal $\hat{g}$. The bound strictly decreases with $\Delta_{\min}$ and remains non-negative for all feasible gaps.*

*Proof.* We begin by applying the goal-conditioned policy difference lemma (Kakade & Langford, 2002), which expresses the suboptimality gap as the expected cumulative advantage under the learned policy:

$$V^*(s_0, \hat{g}) - V^{\pi_\phi}(s_0, \hat{g}) = \mathbb{E}_{\tau \sim \pi_\phi} \left[ \sum_{t=0}^{\infty} \gamma^t A^*(s_t, \hat{g}, a_t) \,\middle|\, s_0 \right], \tag{7}$$

where $A^*(s, g, a) = Q^*(s, g, a) - V^*(s, g)$ and $a_t \sim \pi_\phi(\cdot|s_t, \hat{g})$. Since $V^*(s, g) = \max_{a'} Q^*(s, g, a')$, it follows that:

$$A^*(s, g, a) \leq \max_{a'} Q^*(s, g, a') - Q^*(s, g, a), \tag{8}$$

and therefore:

$$V^*(s_0, \hat{g}) - V^{\pi_\phi}(s_0, \hat{g}) \leq \mathbb{E}_{\tau \sim \pi_\phi} \left[ \sum_{t=0}^{\infty} \gamma^t \left( \max_a Q^*(s_t, \hat{g}, a) - Q^*(s_t, \hat{g}, a_t) \right) \,\middle|\, s_0 \right]. \tag{9}$$

To relate this to the learned Q-function $Q_\theta$, we decompose the expression inside the expectation using the triangle inequality:

$$\max_a Q^*(s_t, \hat{g}, a) - Q^*(s_t, \hat{g}, a_t)$$

$$\leq \left| \max_a Q^*(s_t, \hat{g}, a) - \max_a Q_\theta(s_t, \hat{g}, a) \right| + \left[ \max_a Q_\theta(s_t, \hat{g}, a) - Q_\theta(s_t, \hat{g}, a_t) \right] + \quad (10)$$

$$\left| Q_\theta(s_t, \hat{g}, a_t) - Q^*(s_t, \hat{g}, a_t) \right|. \quad (11)$$

By Assumption 2, the first and third terms are each bounded by $\epsilon_Q$. Summing over the trajectory and taking expectation, their total discounted contribution is:

$$\mathbb{E}_{\tau \sim \pi_\phi} \left[ \sum_{t=0}^{\infty} \gamma^t \cdot 2\epsilon_Q \,\middle|\, s_0 \right] = \frac{2\epsilon_Q}{1 - \gamma}. \quad (12)$$

Next, we consider the middle term, $\max_a Q_\theta(s_t, \hat{g}, a) - Q_\theta(s_t, \hat{g}, a_t)$. Let $\pi_\theta^*(s, g) = \arg\max_a Q_\theta(s, g, a)$ denote the greedy policy with respect to $Q_\theta$. To connect this term to the policy gap $\Delta_{\min}$, we introduce the non-target goal $\bar{g}$ and write:

$$\max_a Q_\theta(s_t, \hat{g}, a) - Q_\theta(s_t, \hat{g}, a_t)$$

$$= \left[ Q_\theta(s_t, \hat{g}, \pi_\theta^*(s_t, \hat{g})) - Q_\theta(s_t, \bar{g}, a_t) \right] + \left[ Q_\theta(s_t, \bar{g}, a_t) - Q_\theta(s_t, \hat{g}, a_t) \right]. \quad (13)$$

The first bracket is bounded in absolute value by $2Q_{\max} = 2R_{\max}/(1 - \gamma)$, due to the boundedness of $Q_\theta$. For the second bracket, we apply Assumption 2 to obtain:

$$Q_\theta(s_t, \bar{g}, a_t) - Q_\theta(s_t, \hat{g}, a_t) \leq \left[ Q^{\pi_\phi}(s_t, \bar{g}, a_t) + \epsilon_Q \right] - \left[ Q^{\pi_\phi}(s_t, \hat{g}, a_t) - \epsilon_Q \right]$$

$$= - \left[ Q^{\pi_\phi}(s_t, \hat{g}, a_t) - Q^{\pi_\phi}(s_t, \bar{g}, a_t) \right] + 2\epsilon_Q. \quad (14)$$

Taking expectation over $s_t \sim d^{\pi_\phi}(\cdot|\hat{g})$ and $a_t \sim \pi_\phi(\cdot|s_t, \hat{g})$, and invoking Assumption 3, we find:

$$\mathbb{E} \left[ Q_\theta(s_t, \bar{g}, a_t) - Q_\theta(s_t, \hat{g}, a_t) \right] \leq -\Delta_{\min} + 2\epsilon_Q. \quad (15)$$

Combining this with the bound on the first bracket, the expected value of the middle term satisfies:

$$\mathbb{E} \left[ \max_a Q_\theta(s_t, \hat{g}, a) - Q_\theta(s_t, \hat{g}, a_t) \right] \leq \frac{2R_{\max}}{1 - \gamma} - \Delta_{\min} + 2\epsilon_Q. \quad (16)$$

Substituting equation 12 and equation 16 into equation 9, and summing the geometric series $\sum_{t=0}^{\infty} \gamma^t = 1/(1 - \gamma)$, we arrive at:

$$V^*(s_0, \hat{g}) - V^{\pi_\phi}(s_0, \hat{g}) \leq \frac{2\epsilon_Q}{1 - \gamma} + \frac{1}{1 - \gamma} \left( \frac{2R_{\max}}{1 - \gamma} - \Delta_{\min} + 2\epsilon_Q \right)$$

$$= \frac{4\epsilon_Q}{1 - \gamma} + \frac{2R_{\max}}{(1 - \gamma)^2} - \frac{\Delta_{\min}}{1 - \gamma}. \quad (17)$$

$\square$

# B  FURTHER ANALYSIS

## B.1  FETCH TASKS

Figure 6 illustrates a comprehensive set of goal-conditioned tasks based on the Fetch environment, designed to evaluate reinforcement learning algorithms in diverse manipulation scenarios. These tasks include: (a) Reach, where the robotic arm moves to a specified target position; (b) Push, involving the displacement of an object to a designated location; (c) PickPlace, requiring the arm to grasp and relocate an object; (d) Slide, where the arm slides an object across a surface; (e) and (f) Push_Obs and Slide_Obs are the tasks that adding some barriers conditioned on Push and Slide tasks; (g) Boxopen, requiring the opening of a box; and (h) Draweropen, involving the opening of a drawer. Each task is conditioned on a specific goal, defined by the desired end-effector position or object state, enabling the assessment of policy performance across a range of complexities and dynamics.

Figure 6: The illustration of Fetch goal-conditioned tasks.

Figure 7: The environment in Point based control tasks.

## B.2 EVALUATION RESULTS ON POINT TASKS

We further evaluate our method on a diverse set of point-based and robotic control tasks, including the environments depicted in Figure 7 (PointCross, PointFourRooms, PointMaze6x6, and PointSpiral7x7). These evaluations compare our approach against the baseline Hindsight Experience Replay (HER) (Andrychowicz et al., 2017), assessing performance through the success rate metric. Figure 8 presents the success rate trends across these tasks. The results, as shown in Figure 8, demonstrate that our method consistently achieves a higher success rate compared to HER across all tasks, with notable improvements observed in complex environments such as PointMaze6x6 and PointSpiral7x7. For instance, our method exhibits a success rate exceeding 0.6 by the 50th training epoch in PointSpiral7x7, surpassing HER's performance of approximately 0.4. Additionally, the enhanced policy gap optimization contributes to greater sample efficiency, as inferred from the steeper learning curves, though detailed sample efficiency metrics are presented in Figures 7 and 8. These findings underscore the efficacy of our approach in capturing underlying environment dynamics and optimizing policy performance across varied goal-conditioned scenarios.

## B.3 VALUE DIFFERENCE MAP IN POINT TASK

To investigate the discriminative capability of learned policies between goal-conditioned and non-target goal action values, value difference maps were constructed by computing the difference between two critic networks respectively in Fig 9. For each spatial position in the four rooms tasks, positive values indicate regions where target-goal-directed actions have higher values, while negative values (red) indicate the opposite.

The experimental results in Fig. 9 demonstrate that BG2RL exhibits significantly larger positive value gaps along critical navigation paths compared to the baseline HER method. Specifically, in the heatmaps, we observe that BG2RL generates larger positive value differences (blue regions) along the optimal trajectories connecting the start positions to the target goals, particularly along the critical paths at the bottom-left and top-right of the map. These blue regions indicate that BG2RL maintains a clear and substantial distinction between the action values of the optimal and suboptimal policies. In contrast, HER displays mixed positive and negative value differences, with some negative values even appearing along the critical paths, suggesting that the optimal and suboptimal policies become more intertwined and harder to distinguish. The progressive increase in mean value

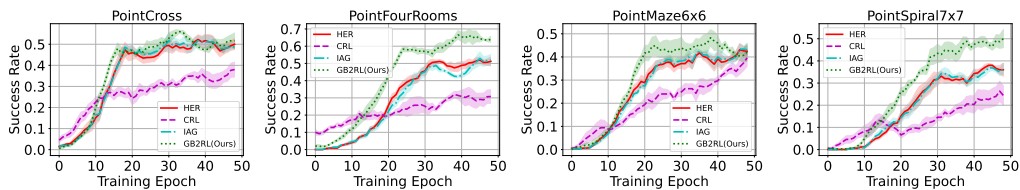

Figure 8: The success rate on Point goal-conditioned tasks.

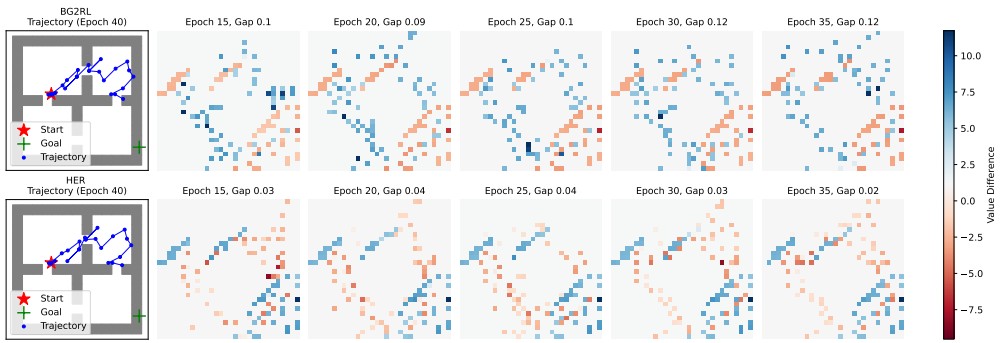

Figure 9: The action value difference map in four rooms goal-conditioned task. BG2RL exhibits larger value gaps along critical path regions compared to HER.

gaps across training epochs—from 0.1 to 0.12 for BG2RL—further highlights that our method successfully enlarges the policy gap over time. In contrast, HER shows relatively smaller and inconsistent gaps (ranging from 0.02 to 0.04), validating the theoretical foundation that a larger action-value gap leads to better goal-directed policy learning and improved task performance.

## B.4 EVALUATION RESULTS ON ANT TASKS

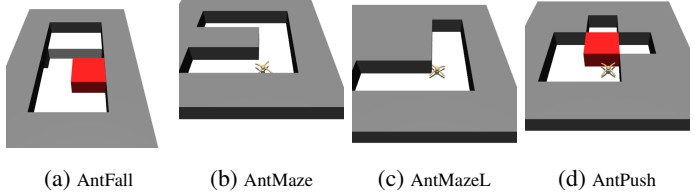

(a) AntFall   (b) AntMaze   (c) AntMazeL   (d) AntPush

Figure 10: The environment in Ant based control tasks.

Figure 11 shows the comparison of our method with other baseline approaches on four Ant-based tasks as Fig. 10: AntFall, AntMaze, AntMazeL, and AntPush. As shown, our method (GB2RL) achieves performance comparable to existing techniques such as HER, CRL, and IAG. The learning curves demonstrate that, while performance varies slightly between tasks, our method consistently performs well, matching or surpassing the baselines in terms of success rate. This illustrates the effectiveness and robustness of our approach in the context of goal-conditioned reinforcement learning for Ant based control tasks.

## B.5 HYPER-PARAMETER INVESTIGATION

Our method utilizes target and non-target goal-conditioned action values, which correspond to the optimal policy we aim to learn for reachable goals and a reference suboptimal policy for unreachable goals, respectively. This approach incorporates a hyper-parameter $\alpha$ that weights the penalization term in the objective function, as defined in Equation 3. Figure 12 presents the success rates of the target policy on Fetch tasks under various $\alpha$ values. The results indicate that the target policy

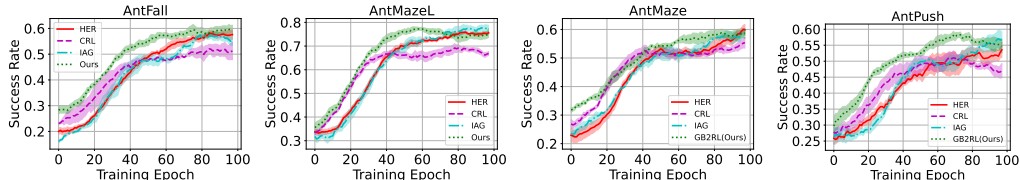

Figure 11: The success rate on Ant based goal-conditioned tasks.

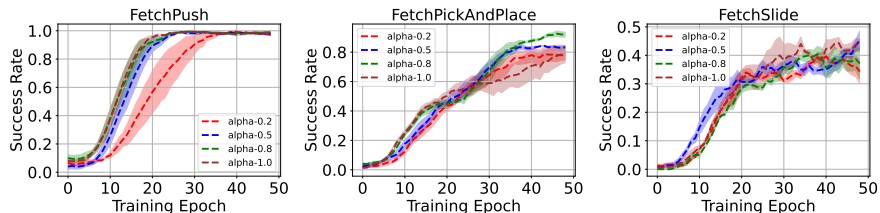

Figure 12: The evaluation results on Fetch tasks with respect to different $\alpha$.

achieves stable and high performance when $\alpha \geq 0.5$, with success rates approaching 1.0 in Fetch-Push and exceeding 0.4 in PickAndPlace and Slide by the 50th epoch for $\alpha = 1.0$. Additionally, we examine the non-target policy's performance in Figure 13, which reveals consistently low success rates across all $\alpha$ settings, confirming that the non-target policy fails to accomplish the tasks effectively during training. These findings validate the effectiveness of the policy gap enhancement, as the target policy converges to superior performance while the non-target remains suboptimal.

### B.6 DECISION PLANE INVESTIGATION

To evaluate the behavioral differences between the proposed BG2RL method and the baseline HER algorithm, policy decision planes are constructed to visualize action magnitude distributions across different goal positions as Fig. 14. For each goal position $(x, y)$ in the grid, the L2-norm of the first three action components output by the respective policies is reported as $z$. The visualization is focused on the X-Y plane because the primary manipulation tasks occur in this horizontal workspace, while the Z-dimension is task-specifically constrained. To further assess policy robustness and generalization capabilities, performance is evaluated on both standard environments and their obstacle-augmented variants.

As shown in Fig. 14, our method BG2RL demonstrates clear advantages over HER in terms of spatially grounded, goal-directed policy learning. In standard tasks (FetchPush and PickAnd-Place), BG2RL generates action distributions that are sharply focused around optimal trajectories—evidenced by concentrated high-magnitude regions—indicating strong discrimination between goal-achieving and non-goal-achieving actions. This aligns with our core design: explicitly maximizing the value gap between optimal and suboptimal policies. In contrast, HER's action maps are more uniform and lack spatial structure, reflecting its tendency to blur the distinction between effective and ineffective behaviors due to insufficient value separation.

Importantly, this advantage becomes even more pronounced in obstacle-augmented environments. While BG2RL adapts its action patterns coherently to navigate around obstacles—preserving clear spatial gradients and goal alignment—HER exhibits increased randomness and degraded structure, suggesting failure to maintain consistent policy discrimination under environmental complexity. Quantitative metrics further validate that BG2RL produces more selective, context-sensitive responses, suppressing irrelevant actions while amplifying goal-critical ones. In summary, this visualization confirms that BG2RL's mechanism of enlarging the action-value gap between optimal and suboptimal behaviors leads to more structured, spatially aware, and robust policies.

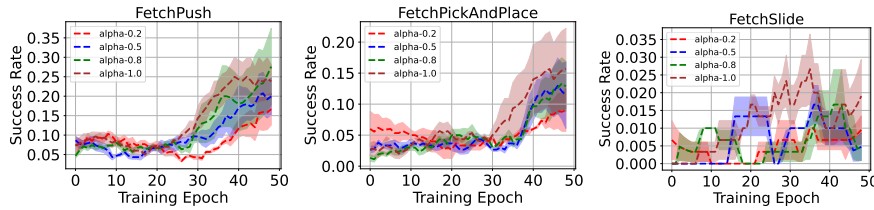

Figure 13: The evaluation results on Fetch tasks with respect to different $\alpha$ for our non-target policy.

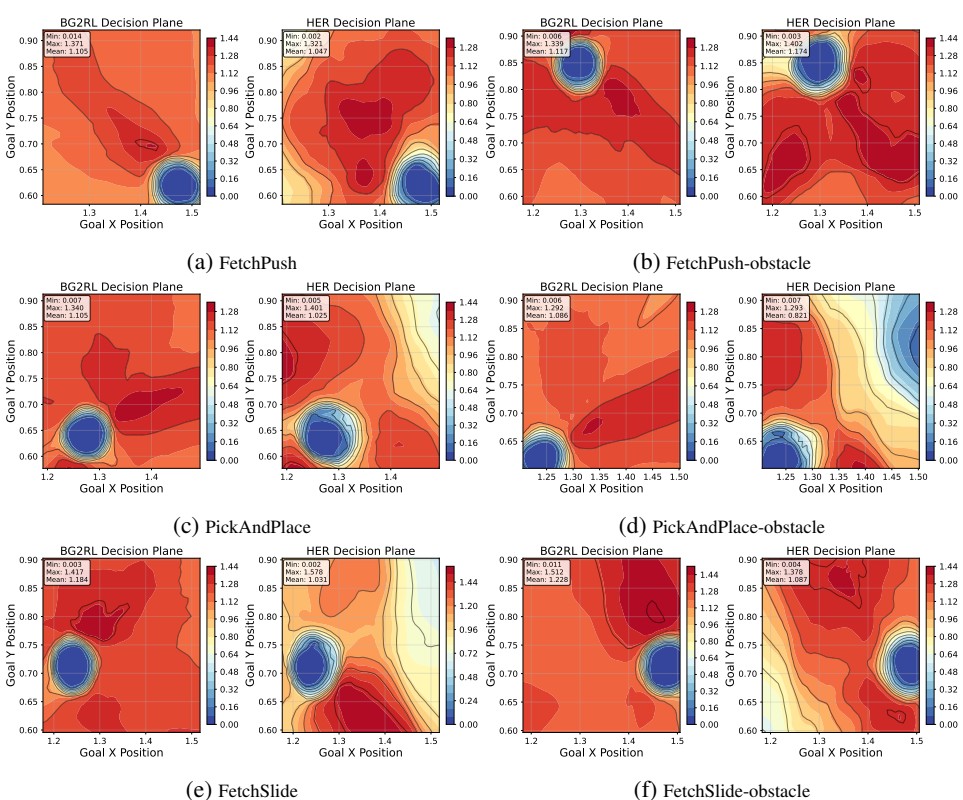

Figure 14: The decision plane of our method and the compared baseline.

### B.7 POLICY GRADIENT FLOW

Fig. 15 illustrates the policy gradient flow for three Fetch-based robotic manipulation tasks, comparing the behaviors of BG2RL and HER across different epochs. The gradient flow is visualized on the XY-plane, where a quiver plot shows action vectors at each grid point, with arrow directions representing the XY components and color intensity indicating action magnitude. The red star marks the goal. To quantitatively evaluate policy quality, we compute the Goal-Directedness Score as shown in the subplot titles, which measures the average alignment between the policy's actions and the direct path to the goal—higher scores reflect more efficient goal-directed behavior.

The visualizations highlight significant differences between BG2RL and HER in terms of goal-reaching behavior. Specifically, BG2RL exhibits a more structured and goal-convergent policy across all environments, with action vectors from various points in the workspace consistently directed towards the goal, forming a coherent "gravitational field" pattern. In contrast, HER often shows divergent or circuitous behavior, especially in regions far from the goal, where action directions become inconsistent and sometimes form vortices or tangential flows. This divergence in HER's policy indicates an inherent ambiguity in distinguishing optimal from suboptimal actions. These visual differences are further validated by the Goal-Directedness Scores presented in Table 2.

Table 2: Goal-Directedness Scores for Fetch tasks.

| Goal-Directedness Scores for Policy Gradient Flow | | | | | |
|---|---|---|---|---|---|
| Epoch | FetchPickAndPlace-v1 | | FetchPush-v1 | | FetchSlide-v1 | |
| | BG2RL | HER | BG2RL | HER | BG2RL | HER |
| 10 | 48.40 | 0.00 | 0.00 | 4.72 | 0.00 | 1.55 |
| 20 | 78.91 | 10.81 | 52.76 | 0.00 | 0.00 | 0.00 |
| 30 | 85.65 | 44.81 | 47.01 | 1.54 | 48.49 | 0.00 |
| 40 | 85.62 | 88.34 | 61.24 | 65.22 | 34.75 | 0.00 |

As shown, BG2RL consistently outperforms HER in goal-directedness across different epochs for all environments. For `FetchPickAndPlace-v1`, BG2RL shows a significant improvement from 48.40 at epoch 10 to 85.62 at epoch 40, while HER remains largely stagnant at 0.00 at epoch 10 and fluctuates around 88.34 at epoch 40. The similar results are also reported in Table 2.

These improvements in BG2RL's goal-directed behavior are a direct result of its contrastive optimization framework, which explicitly maximizes the value gap between target and non-target actions. This method enables BG2RL to learn a more discriminative value function that better distinguishes optimal from suboptimal actions, leading to more stable and direct goal-reaching behaviors.

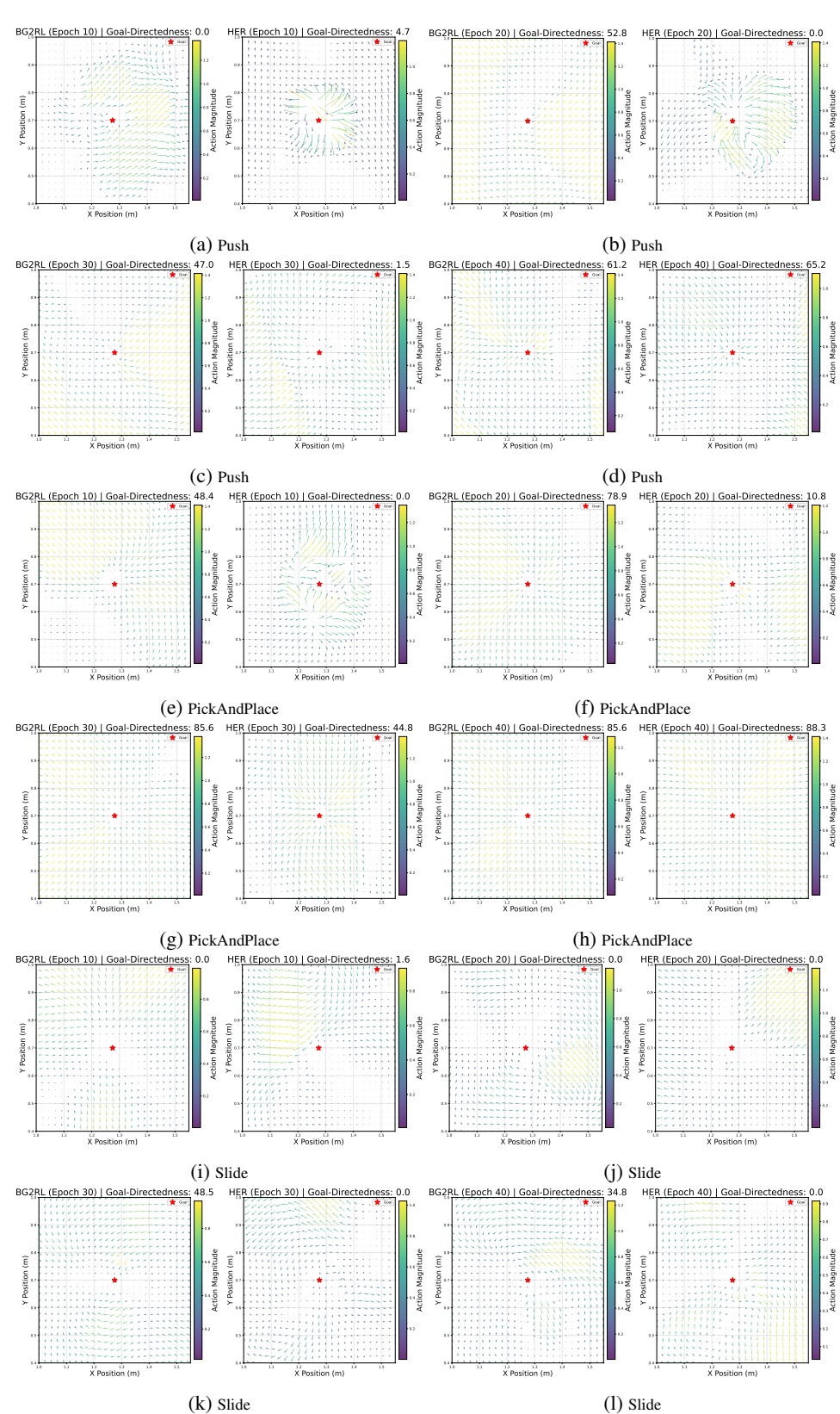

Figure 15: Policy gradient flow visualization in the XY-plane on three goal-conditioned Fetch tasks.

## C  PARAMETERS SETTINGS

Table 3: Parameters for our method

| Parameter | Default Value | Description |
|---|---|---|
| epochs | 50 | Number of training epochs |
| cycles | 50 | Number of sample collection cycles per epoch |
| batch | 40 | Number of times for training model |
| workers | 8 | Number of CPUs for sample collection |
| clip return | 50 | Clip threshold for returns |
| noise | 0.2 | Magnitude of action noise |
| random | 0.3 | Probability of random action exploration |
| buffer size | 1,000,000 | Size of the replay buffer |
| replay | 4 | Ratio of HER replays |
| clip obs | 200 | Clip range for observations |
| batch size | 256 | Batch size for network updates |
| gamma | 0.98 | Discount factor |
| action l2 | 1 | L2 regularization coefficient for actions |
| lr actor | 0.001 | Learning rate for the actor network |
| lr critic | 0.001 | Learning rate for the critic network |
| polyak | 0.95 | Coefficient for soft target network updates |
| test rollouts | 10 | Number of test rollouts |
| clip range | 5 | Clip range for normalization |
| demo length | 20 | Length of demonstration data |
| num rollouts per mpi | 20 | Number of rollouts per MPI process |

Table 3 presents the parameter settings for FetchPush, FetchPickAndPlace, and Fetch-Slide. Additionally, FetchReach uses the settings `n-cycles=5`, `n-batch=1`, and `num-rollouts-per-mpi=1`, with `n-epochs=100` applied uniformly to all Ant-based tasks. For Point tasks, the settings are `n-epochs=50`, `n-cycles=30`, `n-batch=20`, and `num-rollouts-per-mpi=10`.