# OpenReview forum: "Learning to Distinguish: Behavior Gap Optimization for Goal-Conditioned Policy Learning"
_ICLR.cc/2026/Conference — ICLR 2026 Conference Desk Rejected Submission_

### Official Review · Reviewer_6k6U · 2025-10-20

**Soundness:** 2
**Presentation:** 2
**Contribution:** 2
**Rating:** 2
**Confidence:** 3

**Summary:**

This paper presents a novel goal-conditioned RL method, BG2RL, which aims to enhance the performance and robustness of the policy by explicitly increasing the action-value gap between target goals and non-target goals.

**Strengths:**

1. BG2RL has been validated on multiple continuous control tasks, such as Fetch, Point, and Ant, demonstrating its superiority over baseline methods like HER, IAG, and CRL in terms of success rate and sample efficiency.

2. A theoretical analysis (Theorem 1) is provided, proving that increasing the action-value gap can reduce the suboptimality bound of the policy, thus offering a mathematical foundation for the method.

**Weaknesses:**

1.BG2RL combines contrastive learning with goal-conditioned reinforcement learning, a combination that has already been explored in previous works [1,2,3], thus lacking novelty.

2.The BG2RL paper does not discuss the differences from the aforementioned works and appears to adopt a similar approach.

3.The generation mechanism for non-target goals is relatively simple: current non-target goals are sourced from other trajectories, which may not cover all possible failure scenarios. In the future, more intelligent methods for generating negative samples may be required.

4.In highly complex tasks, or when the goal representation is high-dimensional and abstract (e.g., image-based goals), random sampling or simple distance-based sampling may fail to generate "challenging" non-target goals, thereby weakening the effectiveness of contrastive learning.

5.The paper mentions that to apply IAG to continuous action spaces, they generate suboptimal actions by adding clipped random noise to the target policy's actions. This adaptation may severely limit the performance of IAG, as the actions generated by random perturbations could be of very low quality and fail to constitute "competitive suboptimal actions." As a result, the comparison between BG2RL and IAG may not be entirely fair.

6.The potential risks of overestimating or underestimating the value function have not been discussed. The core of the method involves manipulating the value function $Q_{\theta}$. Algorithms like DDPG are inherently susceptible to value function overestimation. By maximizing the gap, BG2RL could amplify any errors in value estimation (whether overestimation or underestimation). The paper does not analyze or discuss this sensitivity.

7.All experiments were conducted in simulation environments, and the performance in the real world remains unknown. I suggest that the authors consider using the environments from OGBench [4] to test their algorithm and compare it with the state-of-the-art methods.

[1] Eysenbach B, Zhang T, Levine S, et al. Contrastive learning as goal-conditioned reinforcement learning[J]. Advances in Neural Information Processing Systems, 2022.

[2] Zheng C, Salakhutdinov R, Eysenbach B. Contrastive difference predictive coding. ICLR, 2024.

[3] Zheng C, Eysenbach B, Walke H, et al. Stabilizing contrastive rl: Techniques for robotic goal reaching from offline data. ICLR, 2024.

[4] Park S, Frans K, Eysenbach B, et al. Ogbench: Benchmarking offline goal-conditioned rl. ICLR, 2025.

**Questions:**

1.Why did you choose older baselines for comparison? Why didn't you include a comparison with the latest online GCRL methods, such as QRL [1] and TD-InfoNCE [2]?

2.Why was there no comparison with CRL [3], especially since it is quite similar to your approach?

3.How does BG2RL perform in high-dimensional environments compared to the aforementioned contrastive GCRL methods?

4.If the assumptions of Theorem 1 are not satisfied, would Theorem 1 still hold in general?

5.If the rewards are a combination of 0 and 1, does BG2RL still remain effective?

6.How does BG2RL perform in real-world environments and environments with image-based observations?

[1] Zheng C, Salakhutdinov R, Eysenbach B. Contrastive difference predictive coding. ICLR, 2024.

[2] Wang, Tongzhou, et al. "Optimal goal-reaching reinforcement learning via quasimetric learning." ICML, 2023.

[3] Eysenbach B, Zhang T, Levine S, et al. Contrastive learning as goal-conditioned reinforcement learning[J]. Advances in Neural Information Processing Systems, 2022.

---

### Official Review · Reviewer_z5af · 2025-10-28

**Soundness:** 2
**Presentation:** 2
**Contribution:** 3
**Rating:** 4
**Confidence:** 3

**Summary:**

The paper focuses on goal-conditioned RL and aims to enlarge the value estimation gap between optimal and suboptimal actions. Specifically, it samples target goals from adjacent states within the same trajectory and non-target goals from the states in other trajectories whose value estimates are closest to the target ones. The policy is then trained to maximize the values for target goals while conditionally minimizing the value for non-target goals. Experiments in several continuous control environments demonstrate that the proposed method achieves better performance than other baselines.

**Strengths:**

- The overall algorithmic pipeline is clear and easy to follow.

- The idea of explicitly shaping the value gap is intuitive and potentially useful

**Weaknesses:**

- The paper’s explanations are sometimes unclear, making it difficult to fully understand key concepts (see questions 1-2).

- Some experimental results and corresponding discussions do not align well (see questions 3-5).

- In Theorem 1, the second term $\frac{2R_{max}}{(1-\gamma)^2}$ is a very large constant, which makes the bound quite loose. It is also unclear under what conditions this bound would converge to 0. Furthermore, the distribution of non-target goals $\overline g$ in Theorem 1 is not clearly given.

- The choice and influence of the important parameter $\delta$ are not clearly explained and empirically studied. Beside, It should include an ablation on the non-target loss term (e.g., let $\alpha=0$).

- Although the paper aims to enlarge the value gap between optimal and suboptimal actions, the method actually increases the difference between correct goals and generated goals. The connection between these two intuitions is a bit unclear.

**Questions:**

1. In Section 4.1, the phrase “minimize the Q-value” to “encourages the agent to explore different goals” feels abrupt and confusing, though it is clarified later in Section 4.2.

2. In Section 4.4, I am sure why we need to "amplifies the gap" when “it correctly assigns lower values to target goals” and what “ordering is reversed” specifically mean.

3. In Section 5.1, the statement “CRL attains higher success rates than HER in Ant- and Point-based tasks but lower rates in Fetch-based tasks” does not seem to be supported by the figures. It is hard to identify the advantage of CRL from Figures 8 and 9.

4. In Section 5.2, the claim that “HER never reaches the level of BG2RL” appears inconsistent with Figures 3(d–f), where HER seems to reach a similar level.

5. It is not immediately clear how the results support the statement “The IAG surface is overly flat” from Figure 4.

6. In Figure 5, it is a bit hard to see how “ALL results in unstable performance in FetchPickAndPlace and FetchSlide.”

7. The results for binplace and boxopen show that other baselines perform significantly worse. It would better if the authors could discuss possible reasons for this.

8. In Section 5.4, one of the formulas is missing a right bracket.

---

### Official Review · Reviewer_Wu7J · 2025-10-31

**Soundness:** 2
**Presentation:** 2
**Contribution:** 2
**Rating:** 4
**Confidence:** 3

**Summary:**

The paper introduces a goal-conditioned reinforcement learning method that tries to make the value estimates for success-leading actions clearly larger than those for failure-leading ones. It constructs two kinds of goals for each state: target goals taken from reachable future states within the same trajectory, and non-target goals mined from other trajectories that are hard to distinguish from the target. The actor is trained to increase the gap between the target and non-target action values, with a gating term that avoids over-penalizing when the ordering is already correct. The authors present a suboptimality bound that decreases with the learned gap, and report higher success rates than HER, IAG, and a contrastive RL baseline on Fetch-style manipulation tasks and obstacle variants.

**Strengths:**

* Conceptually, focusing training on separating target from confusing non-target goals is a neat way to combat value over-lap that often hurts goal-conditioned policies. The paper gives a clear high-level motivation, a simple training objective with an indicator gate, and an attempt at theory connecting larger learned gaps to smaller suboptimality.

* Empirically, on several MuJoCo Fetch environments and obstacle variants, the method reaches higher or more stable success rates than HER, an adapted IAG, and a contrastive baseline, and includes an ablation indicating that unconditional suppression of non-target values can destabilize learning.

**Weaknesses:**

Despite the appealing idea, several technical choices and explanations are confusing or under-supported.

First, the target-goal selection appears inconsistent. The text states that in the sparse negative reward setting, higher action values correspond to shorter paths, yet it then selects the goal with the minimum action value as the most efficient target, which contradicts the earlier explanation. This makes it unclear what is actually being optimized when choosing targets. Clarifying the sign conventions and the rationale for selecting minima versus maxima is necessary.

Second, the non-target goal is picked from other trajectories by minimizing the absolute gap to the target value. Calling these goals “unreachable” is misleading, since they are reachable under some behavior, and the selection relies directly on the current critic. This creates a circular dependency and risks reinforcing early critic biases. There is no analysis of false negatives, nor diagnostics on how often the indicator gate fires during training.

Third, the theory is stated at a high level with strong assumptions, including a uniform Q-approximation error and a strictly positive expected gap. The bound subtracts the learned gap term, but there is no practical procedure that certifies the assumptions, nor empirical measurement linking the bound components to observed performance. The presentation does not make clear how the theoretical quantities map to the implemented indicator-gated loss.

Fourth, the experimental comparison set is dated and limited for ICLR 2026 standards. Using DDPG as the backbone and comparing mainly to HER, an adapted IAG, and one contrastive baseline leaves out widely used and stronger goal-conditioned baselines such as TD3+HER and SAC+HER, as well as recent goal-conditioned Q-learning and diffusion-based actor-critic variants. Claims of robustness would be more convincing with these stronger baselines and on harder long-horizon or visual tasks. The paper also does not detail environment interaction budgets per epoch, so sample efficiency across methods is hard to judge.

Fifth, several presentation issues hinder clarity. Figures use small fonts and inconsistent labeling, and the narrative around Figure 1 mixes example numbers with informal statements, which contributed to the confusion about target-goal selection. Some terminology is imprecise, for example “unreachable” versus “non-target,” and “policy gap” versus “action-value gap.”

**Questions:**

* Target goal selection: given the explanation that higher action values imply shorter paths in this reward convention, why does the algorithm pick the goal with the minimum action value as the target? Please reconcile this inconsistency, or correct the description, and report results for both choices. A small controlled study would help.

* Negative sampling semantics: in what precise sense are non-target goals “unreachable” from the current state with the current goal, and how often do they lead to success if executed by the current policy? Please provide statistics over training, including how frequently the indicator activates and the empirical distribution of target versus non-target values.

* Sensitivity to hyperparameters and design: how sensitive is performance to the indicator gate, its coefficient, the distance threshold for target sampling, and the number of trajectories searched for negatives? Please include sweeps and failure cases.

* Stronger baselines: can you add results with TD3+HER and SAC+HER on all tasks, as well as one recent contrastive or diffusion-based goal-conditioned method, using matched budgets and reporting environment steps? Without these, it is difficult to gauge novelty and impact.

* Theory to practice: can you measure the learned gap term over time, relate it to the bound, and show whether larger learned gaps predict higher success rates across seeds and tasks? This would better justify the theoretical claim.

---

### Official Review · Reviewer_GbVL · 2025-11-01

**Soundness:** 2
**Presentation:** 2
**Contribution:** 1
**Rating:** 2
**Confidence:** 4

**Summary:**

The paper proposes a method for enlarging the separation between action values conditioned on target and non-target goals. Specifically, it employs a heuristic rule to select target and non-target goals from different rollouts, based on which an optimization objective is designed. The method was validated in relatively simple experimental scenarios, where it demonstrated performance improvements over several classical baselines.

**Strengths:**

The paper provides a relatively detailed description of the proposed method.

**Weaknesses:**

1. The paper's core motivation, "this bootstrapping process can propagate and amplify estimation errors when the value functions for different goals are similar", lacks experimental validation.

2. The goal selection mechanism is inherently heuristic. Furthermore, as suggested by Equation (3), the method's effect is most prominent in the early stages of training. However, during this phase, the Q-function is poorly estimated and contains significant noise. This raises concerns about the reliability of selecting target and non-target goals based on such a noisy Q-function.

3. The method selects target goals for a given state from within a local delta-ball. This means the constructed positive and negative goal pairs are always in the immediate vicinity of the current state. It is questionable whether this local strategy can effectively solve long-horizon tasks, where the desired goal is far from the initial state. The current experimental setup, with a maximum episode length of only 50 steps, consists of relatively simple GCRL tasks that do not adequately test this capability.

4. The paper states that the rollouts for selecting non-target goals differ from those for target goals only in their initial states. This implies that the initial state distribution is crucial for the successful selection of informative non-target goals. The authors should clarify what characteristics of an initial state distribution are beneficial for this process and discuss the method's sensitivity to it.

5. The experiments compare against HER. It is important to clarify whether the implementation of BG2RL also incorporates HER's replay goal mechanism.

6. (a) What environment suite was used for the experiments (e.g., Gymnasium, DeepMind Control)? This should be clearly specified. (b) The training procedure seems to be extremely data-limited. With only 50 epochs and a maximum episode length of 50 steps, does this mean the entire training process for each algorithm involves only 2,500 environment samples?

7. Section 5.3's focus on multi-goal tasks implies that the preceding experiments (Sections 5.1 and 5.2) use a single, fixed desired goal per episode. If this is the case, it represents a significant simplification. In fact, the described setting seems even simpler than standard **Reach** tasks. The GCRL community typically evaluates methods in multi-goal settings, where the desired goal is sampled from a distribution at the beginning of each episode. The authors should clarify the goal-setting for all experiments and justify why a simplified single-goal setting was used.

8. The conclusion in lines 433-435 feels abrupt. How does this specific conclusion relate to the paper's central argument? In other words, what is the explicit connection between the stated objective of "maximizing the action value gap between target and non-target goals" and this particular finding?

9. There are several typos, (1) In Figure 2 and Table 1, "BG2RL" is misspelled as "GB2RL". (2) The y-axis labels in Figures 3(d-e) seem to be incorrect. (3) The equation on line 459 is missing a closing parenthesis.

**Questions:**

refer to weaknesses

---

### Note · Program_Chairs · 2026-01-17
**Submission Desk Rejected by Program Chairs**

The following references in this submission do not refer to real documents and/or have major errors in bibliographic information:

 Bogdan Mazoure Zheng, Kishan Thakkar, Samuel Goldberg, Daniel Hudson, Jesse Farebrother, Will Dabney, Aaron Courville, and Marc G Bellemare. Contrastive value learning: Implicit models for simple offline rl. In International Conference on Machine Learning, pp. 42169-42185, 2023.
Jesse Wang, Yunfei Jiang, Junfeng Zhu, Yi Ma, and Pulkit Agarwal. Proto-value networks: Scaling representation learning with auxiliary tasks. In International Conference on Learning Representations, 2023.